# Effectiveness of Remote Interventions to Improve Medication Adherence in Patients after Stroke: A Systematic Literature Review and Meta-Analysis

**DOI:** 10.3390/bs13030246

**Published:** 2023-03-11

**Authors:** Yan Yee Cherizza Choi, Micah Fineberg, Aikaterini Kassavou

**Affiliations:** 1Department of Public Health and Primary Care, Clinical Medical School, University of Cambridge, Cambridge CB2 0SR, UK; 2UCL Queen Square Institute of Neurology, University College London, London NW3 2PF, UK

**Keywords:** medication adherence, stroke patients, telemedicine, mHealth

## Abstract

Background: Stroke affects more than 30 million people every year, but only two-thirds of patients comply with prescribed medication, leading to high stroke recurrence rates. Digital technologies can facilitate interventions to support treatment adherence. Purpose: This study evaluates the effectiveness of remote interventions and their mechanisms of action in supporting medication adherence after stroke. Methods: PubMed, MEDLINE via Ovid, Cochrane CENTRAL, the Web of Science, SCOPUS, and PsycINFO were searched, and meta-analysis was performed using the Review Manager Tool. Intervention content analysis was conducted based on the COM-B model. Results: Ten eligible studies were included in the review and meta-analysis. The evidence suggested that patients who received remote interventions had significantly better medication adherence (SMD 0.49, 95% CI [0.04, 0.93], and *p* = 0.03) compared to those who received the usual care. The adherence ratio also indicated the interventions’ effectiveness (odds ratio 1.30, 95% CI [0.55, 3.10], and *p* = 0.55). The systolic and diastolic blood pressure (MD −3.73 and 95% CI [−5.35, −2.10])/(MD −2.16 and 95% CI [−3.09, −1.22]) and cholesterol levels (MD −0.36 and 95% CI [−0.52, −0.20]) were significantly improved in the intervention group compared to the control. Further behavioural analysis demonstrated that enhancing the capability within the COM-B model had the largest impact in supporting improvements in adherence behaviour and relevant clinical outcomes. Patients’ satisfaction and the interventions’ usability were both high, suggesting the interventions’ acceptability. Conclusion: Telemedicine and mHealth interventions are effective in improving medication adherence and clinical indicators in stroke patients. Future studies could usefully investigate the effectiveness and cost-effectiveness of theory-based and remotely delivered interventions as an adjunct to stroke rehabilitation programmers.

## 1. Introduction

Stroke is a cerebrovascular disease that refers to the disturbance of cerebral functions due to a restricted supply of blood to the brain [1]. According to the World Health Organisation (WHO), stroke accounts for approximately 15 million incident cases globally every year, attributing to 5 million deaths and another 5 million chronic disabilities, such as memory problems or speech disorders [2,3]. The absolute incident and prevalent numbers of strokes increased by 70% and 85%, respectively, between 1990 to 2019, and the numbers are expected to keep rising due to the aging population [4,5]. This can lead to significant global health burdens due to increased health and social care demands, as well as depleted productivity [6]. Thus, follow-up care for stroke is essential for secondary prevention. Evidence suggests that the risks of further stroke episodes are significantly higher, with an increased cumulative risk of 26.4% by five years following the initial stroke and 39.2% by ten [7]. Recurring strokes also present higher mortality rates than initial strokes [8]. One way to control the risks associated with recurrent stroke is to support adherence to pharmacological treatment such as antiplatelets and anticoagulants to prevent the further formation of clots and/or medications to control patients’ blood pressure and cholesterol levels [9]. A previous study found that an average discharged stroke survivor is prescribed 11 medications from 5.4 different drug classifications, including antihypertensives, antiplatelets, antianginals, and antidepressants [10]. Medication non-adherence in patients with long-term health conditions is only approximately 50% in high-income countries and even lower in low-income countries, accounting for over 100,000 global deaths and $100 billion in medical costs per year [7,8]. Moreover, only two-thirds of stroke patients were estimated to comply with their prescribed medications [11]. Medication non-adherence limits treatment efficacy and its benefits, resulting in adverse clinical outcomes and increased healthcare expenditure [12].

Behavioural interventions are strategies that alter the trajectory and lower the risks of recurrent stroke through supporting behaviour change that increases adherence prescribed to treatments. Examples include behavioural therapy, reminders, dosage simplification, and the dissemination of information regarding the importance of adherence. Emerging intervention strategies are further augmented with digital technology that facilitates resource efficiency, enabling widespread use and reducing health inequalities [13]. These approaches enable patients to access expert advice remotely using technology such as phone or video calls or SMS text messages. For instance, telemedicine entails remote physician-patient consultations using telecommunications, including phone or video calls, and mobile health (mHealth) technology refers to the utilisation of mobile devices to improve consultations and patient access to healthcare advice and support [14,15]. With most households having access to internet or owning at least one mobile device, the target audience of telemedicine and mHealth interventions is significantly wider and bigger than in-person consultations [16]. Since the COVID-19 pandemic outbreak, the majority of healthcare consultations have shifted to remote-based consultations to protect bothpatients and health care professionals [17]. The use of combined telehealth and mHealth interventions have been pivotal in facilitating continuous care during the pandemic, suggesting potential capacity to accommodate efficient service provision [17]. Furthermore, remote interventions are potentially more cost-effective than current usual care service as they overcome geographical barriers and save clinical, administrative, and consultation time and resources [18]. 

Previous reviews have demonstrated the potential of digital interventions in improving medication adherence in patients with non-communicable diseases, such as cardiovascular diseases and stroke [5,19,20,21,22,23]. However, research is sparse, and a wide range of challenges remain unanswered, such as a limited understanding of the intervention mechanisms of action due to their complex multi-level and multi-component nature [5]. Improving our apprehension of their mechanisms of impact is crucial for informing future developments of effective and replicable ways to support behaviour changes, tackle medication adherence issues, and enhance the clinical effectiveness of the interventions to tackle medication adherence issues. To the best of our knowledge, there is no review that has identified the intervention mechanisms that support adherence to pharmacological treatments in stroke patients.

This review aims to provide evidence on the intervention mechanisms of action based on the Capability Opportunity Motivation-Behaviour (COM-B) model. The COM-B model is a widely accepted behavioural theory that operationalises how intervention strategies target and modify underlying influences of behaviours to bring about desired effects [24]. The model recognises three overarching influences of the behaviour, namely capability, opportunity, and motivation [24]. Capability is defined by the individual’s psychological and physical abilities, such as their knowledge to participate in an activity. For example, interventions could aim to enhance capability by providing accessible memory aids and improving patients’ knowledge and understanding of the disease and its treatment via the provision of information. Opportunity refers to external contextual factors that enable the behaviour to occur; for example, by improving access to medication or addressing economic issues by adjusting medication costs to improve its affordability. Finally, motivation refers to the internal processes that also affect an individual’s decision-making, such as appraisals on the perceived health benefits of the prescribed treatments, anticipatory beliefs about potential side effects, or beliefs about the necessity to take medications due to potential reduced observable symptoms or anticipated medication effects [24]. 

This review aims to assess the clinical impacts of telemedicine and mHealth interventions in improving medication adherence in stroke patients, including phone and video call consultations with a healthcare provider, automated phone calls and SMS text messages, and mobile applications. It also addresses the literature gaps by exploring the COM-B intervention functions to explain the clinical effectiveness and acceptability of the received interventions. Overall, this review will generate new and replicable knowledge on the effectiveness and the mechanisms of action of remote interventions that prompt clinically meaningful changes in stroke patients. This can, in turn, provide evidence-based references for the development of future remote clinical interventions and recommendations for improving current, usual care treatments.

## 2. Materials and Methods

A review protocol was developed and registered with PROSPERO (Registration Number: CRD42022322603) [25]. There were no deviations from the protocol. The study methods and results are reported following the PRISMA 2020 Guidelines [26].

### 2.1. Study Eligibility

The PICOS framework was applied to define the inclusion criteria for this review. Included studies must explicitly report that the majority of patients (>50%) were diagnosed with stroke(s), including ischaemic, haemorrhagic, and/or transient ischaemic. Included patients must be over 18 years old and on prescribed pharmacotherapy for stroke [27]. Eligible interventions should be facilitated remotely to support medication adherence in stroke patients. The intervention content could consist of any behavioural change strategies such as goal setting or education to improve treatment adherence behaviours by modifying capability, opportunity, or motivation. Studies that explored the effects of remote interventions on medication adherence in non-stroke patients (e.g., hypertension and diabetes) were excluded. Studies that consisted of interventions with the majority of behavioural advice facilitated by a healthcare provider in-person, such as patients being signposted to an educational material during a general practitioner visit, were excluded. Usual care with in-person or minimal interventions was used as a comparator. The primary outcome was changes in medication adherence, and secondary outcomes included (a) changes in clinical indicators (lipid profile and/or blood pressure levels); (b) evidence of an association between the COM-B intervention components of the behavioural change interventions and potential intervention effectiveness; and (c) patients’ acceptability of their received intervention.

### 2.2. Search Strategy

A systematic search of the electronic databases PubMed, MEDLINE via Ovid, the Cochrane CENTRAL, Web of Science, SCOPUS, and PsycINFO was conducted in January 2022 to identify relevant eligible studies for this review. The search strategy was developed with keywords based on the PICOS framework guidelines and other relevant reviews, resulting in free-text search terms and Medical Subject Headings (MeSH terms) (Appendix A). The search terms were initially developed in MEDLINE via Ovid to identify studies that contained keywords from each category, including stroke (#1), intervention category (#2), the primary outcome (#3), and randomised controlled trial (RCT) (#4). Search filters for the RCTs, humans, and the English language were applied. All studies that were published before January 2022 were reviewed. Grey literature was not searched.

### 2.3. Study Screening

Study records obtained from the six databases were imported into the systematic review web application Rayyan. Duplicate studies were removed according to the titles and authors, whereby the most comprehensive version of each study was retained. Afterward, studies were independently screened by three reviewers (CC, MF, and AK) against the PICO criteria for inclusion and exclusion, and disagreements were resolved with discussion among the three reviewers until a consensus was reached. References of the included studies were also screened for inclusion in any additional studies. Authors of inaccessible studies or protocols only were contacted via email for access to full texts, and the studies were excluded if there was no satisfactory response from the authors. Study protocols with or without a statistical analysis plan were excluded at this stage. Studies that did not meet the inclusion criteria were excluded for reasons marked in Rayyan. Eligible studies were put forward for data extraction.

### 2.4. Data Extraction and Coding

Outcome data were extracted into the software Review Manager *5.4.1* (Revman 5.4.1) [28], and the characteristics of each study were summarised with an Excel data extraction form. All available outcome data were extracted from the eligible studies and explored for further analysis.

Medication adherence and clinical outcome data from intervention and comparator groups were extracted with information on the time point from baselines, measurement methods, and their units of measurement. Objective and subjective measures were both reported, but objective data were selected if a study included both measurements to reduce bias. When outcomes were measured using more than one methods in the same study, the more comparable measure across the studies was selected. Assuming that changes in the data value increase with time following the termination of the study, values that were obtained from the latest in the trials were selected as the most representative [29]. Intention-to-treat analysis data were used if available; otherwise, available cases were selected. Unadjusted baselines and follow-up data were extracted where possible [30]. Changes from baselines were extracted or calculated with raw data; otherwise, only follow-up data were extracted if the baseline values were not reported in the primary studies. Any estimated data that were not aligned with the reported values were discussed among the reviewers to decide if they should be included in the analysis, and reviewers reached a consensus through discussion and made the final decisions. 

The intervention components were coded based on the COM-B model, which included capability, opportunity, and motivation. The reviewers coded intervention strategies and mapped them against each of the COM-B components to describe the intervention mechanisms of action. Application of the COM-B model allowed analysis of the overarching intervention functions when there were multiple approaches reported in the same study or across studies. 

### 2.5. Risk of Bias in Individual Studies

The risk of bias in each study was assessed based on the Cochrane Risk of Bias tool, Version 2 (RoB 2) [31]. Specifically, the primary outcome, or changes in medication adherence, was selected for the RoB v2 evaluation unless unavailable. Otherwise, the secondary outcome, or changes in systolic blood pressure, was selected. Two researchers completed the evaluations independently and discussed resolving any disagreements.

### 2.6. Meta-Analysis

Revman 5.4.1 was used to conduct the meta-analysis. The results of individual RCTs included in this review were summarised into weighted overall effect estimates, which were utilised to summarise the effect, magnitude, and statistical significance of the interventions. 

The standardised mean difference (SMD) was deemed the appropriate summary statistics for the continuous data of the primary outcome (i.e., medication adherence) due to the varied outcome measures, units, and scales extracted from individual studies. The mean difference (MD) was used to summarise clinical indicators, which were measured on the same scale and unit between studies. The pooled standard deviation (SD) was computed as the effect size statistics. Binary data for adherence and clinical indicators (adherent or total and achieved controlled BP levels or not) were used for dichotomous outcomes and summarised using odds ratios.

A fixed-effects model was used for clinical outcomes, and a random-effects model was used for medication adherence outcomes as the latter accounts for heterogeneity in true effect sizes between studies, which aligns with the nature of the studies included in this review that estimate varying yet similar intervention effects [32]. The threshold values of I^2^ statistics between 0 and 40% were interpreted as “might not be important”, 30 and 60% as “moderate heterogeneity”, 50 and 90% as “substantial heterogeneity”, and 75 and 100% as “considerable heterogeneity” [31]. Results with moderate to high heterogeneity were further investigated to explore potential mediators by conducting quantitative subgroup analyses.

### 2.7. Subgroup Analysis

Two quantitative subgroup analyses were conducted to explore the heterogeneity caused by potentially varying intervention components or outcome measurements, including (a) COM-B components and (b) the method used to estimate potential changes in medication adherence. 

### 2.8. Analysis of Intervention Acceptability

To estimate the intervention acceptability and explore its impact on influencing usage and adaptation of the intervention strategies, we developed a coding framework with two overarching categories, including and evaluating perceptual and practical components of the interventions. We hypothesised that these two components predominantly impact intervention acceptability. The majority of the studies included in this review determined acceptability by measuring satisfaction (perceptual) and usability (practical abilities) using structured or open-ended questionnaires and in-depth interviews. The data were first mapped under the two themes and then were critically appraised with the qualitative evidence supplementing the descriptions of the quantitative data. 

## 3. Results

A total of 1849 studies were identified from the searches. Duplicate studies were removed on *Rayyan,* and 1673 studies remained for the title and abstract screening. Afterward, 1610 studies were excluded following the initial screening based on our pre-specified criteria. The reasons for exclusion are stated in Figure 1. The remaining 63 studies were considered for full-text screening, in which the proportion of agreement between the independent reviewers was 71.4%. After discussion, 10 studies were included in the quantitative meta-analysis [15,33,34,35,36,37,38,39,40,41]. Grey literature was not searched, and no additional studies were included from searching the references of the eligible studies. 

All ten included studies were RCTs with remote interventions that supported treatment adherence by modifying one or more of the COM-B model components, which was absent in the comparator groups. A total of 3323 participants were randomised into intervention or comparator groups. Two studies measured medication adherence using clinical indicator biomarkers, but since the remote interventions were designed to primarily improve medication adherence in stroke patients, we deemed the results sufficient for evaluating the effectiveness of the intervention and its behavioural components. 

Among the ten studies, three were implemented in Pakistan, two in China, two in the USA, and one study each in the UK, Denmark, and Sweden. The duration of the studies ranged from 90 days to 36 months (Appendix A). All studies conveyed remote interventions, in which seven studies had interventions delivered by telephone, three by SMS text messages, two by mobile applications, one by video calls, and one by email. The most frequently coded COM-B intervention component was “Capability”, which was coded in six studies, followed by “Motivation” and “Opportunity” each coded in four studies (Appendix A). None of these behavioural components were coded in the comparator groups. In total, five studies explored patients’ acceptability of their received interventions.

### 3.1. Risk of Bias Assessment

The Cochrane Risk of Bias version 2 (RoB v2) tool was used to assess the risk of bias in the included studies, which entail nine individual RCTs and one cluster RCT. The results for the individual RCTs assessment are shown in Figure 2, in which two studies had an overall high risk of bias due to deviations from the intended intervention and missing outcomes, respectively, five studies had some concerns of risk, and two had a low risk of bias. The most unclear risk of bias was due to subjective measurements of the medication adherence outcomes. The cluster randomised trial had an overall high risk of bias due to risk introduced by the recruitment process and some concerns of bias in the measurement of the outcome.

### 3.2. Meta-Analysis

#### 3.2.1. Medication Adherence

Meta-analyses for changes in medication adherence behaviours were conducted using both continuous and dichotomous outcomes (Figure 3). The continuous outcome analysis showed that the intervention supported moderate improvements in medication adherence behaviours compared to the control (SMD = 0.49, 95% CI [0.04, 0.93], and *p* = 0.03). However, the heterogeneity was high and statistically significant (Tau^2^ = 0.30, Chi^2^ = 43.53, df = 6, *p* < 0.00001, and I^2^ = 86%), suggesting varying intervention effects among the studies. The dichotomous outcome analysis estimated a 30% increase in improvements in behaviour change for the intervention compared to the control (OR = 1.30, 95% CI [0.55, 3.10], and *p* = 0.55). Specifically, the study by Yan et al. [40] contributed most of the overall observed effect, suggesting statistically significant improvements of up to 37% in the intervention compared to the control. The difference between the two groups was non-statistically significant, and the heterogeneity was high (Tau^2^ = 0.37, Chi^2^ = 8.51, df = 2, *p* < 0.01, and I^2^ = 76%). Overall, despite the results being non-statistically significant for the dichotomous outcomes, there appears to be a positive effect of remote interventions on improving medication adherence behaviours in stroke populations.

#### 3.2.2. Systolic Blood Pressure

Meta-analyses of changes in systolic blood pressure (sBP) revealed a significant mean change of −3.73 mmHg (95% CI [−5.35, −2.10] and *p* < 0.00001) in the intervention group compared to the control, suggesting potentially clinically meaningful effects of the interventions in improving the sBP (Figure 4). Further test suggested low heterogeneity (26%). For the dichotomous outcomes, the participants who received the interventions were three times more likely to achieve the threshold of sBP control level (OR = 2.89, 95% CI [2.12, 3.94]), and *p* < 0.00001) (Figure 4) compared to the comparator group. There was no heterogeneity detected between the studies (I^2^ = 0%). In both cases, there appear to be statistically significant and clinically important effects of remote interventions on decreasing sBP. 

#### 3.2.3. Diastolic Blood Pressure

Similar to sBP, the intervention group had a significant reduction in diastolic blood pressure (dBP) compared to the control (MD = −2.16, 95% CI [−3.09, −1.22], and *p* < 0.00001), and the participants who received the interventions were 2.5 times as likely to achieve threshold of dBP control values (OR = 2.45, 95% CI [1.64, 3.67], and *p* < 0.0001) (Figure 5) than the comparator group. The analysis provided statistically significant evidence of the intervention effects on reducing the dBP, and the heterogeneity was none to moderate.

#### 3.2.4. Low-Density Lipoprotein Cholesterol

Low-density lipoprotein cholesterol (LDL-C) levels were significantly reduced in the intervention group compared to the control (MD = −0.36, 95% CI [−0.52, −0.20], and *p* < 0.0001) (Figure 6). Substantial heterogeneity was found between the studies contributing to the observed effect (I^2^ = 62%). For the dichotomous outcomes, the participants who received the intervention were twice as likely to achieve LDL-C control than those in the control group (OR = 2.27, 95% CI [1.69, 3.05], and *p* < 0.00001) (Figure 6). No heterogeneity was found (I^2^ = 0). 

### 3.3. Subgroup Analysis of Intervention Behavioural Components

Quantitative sub-group analysis of the intervention mechanisms of action recommended that supporting patients’ capability was most effective in enhancing medication adherence behaviours in stroke patients (SMD = 0.54 and 95% CI [−0.18, 1.26]), followed by motivation (SMD = 0.42 and 95% CI [−0.17, 1.01]), and opportunity (SMD = 0.25 and 95% CI [−0.26, 0.75]) (Appendix A). For instance, interventions that reminded participants to take their medications at the prescribed time and dosage seemed to have enhanced patients’ psychological capability to perform the behaviour in real-time and thus improve adherence to the prescribed treatments. However, high heterogeneity was found within each of the COM subgroups, suggesting that various, or a combination of behaviour change, strategies modified the underpinning of the behaviour change, or the delivery modes and the adoption of the change, and supported improvements in medication adherence.

Additional subgroup analyses compared the results between the self-reported and objective measurements, in which no significant difference was found. This provides confidence in the validity of the measured intervention effects (Appendix A). 

### 3.4. Subgroup Analysis of Intervention Acceptability

Critical appraisal of the data was performed on five studies to investigate patients’ acceptability of their received remote interventions. Among those studies, three investigated the patients’ acceptability based on percentages and quantitative scales, and three explored elements that supported intervention usability and satisfaction through open-ended interviews and questionnaires, exploring parameters such as the relevance of the information received and the digital literacy effort required (Appendix A).

The evidence synthesis indicated high intervention acceptability. Specifically, most evidence suggested high satisfaction, as the majority of participants provided positive feedback and stated that the received interventions supported adherence to their prescription. Quantitatively, this was translated to satisfaction and usability scores of no less than 80%. Answers to open-ended and in-depth questions also showed patients’ confidence in accessing, using, or implementing intervention strategies, as they described the interventions as “useful”, “informative”, and “convenient”. Unstructured feedback on the practical challenges from one study indicated that the patients liked the “automatic transmission of data”, but some negative feedback included technical issues such as their phone running out of battery or confusion on how to use the phone functionalities. Overall, no privacy breach or concerns was recorded. One study also asked for patients’ recommendations and suggestions to increase intervention accessibility, and the majority of the responses indicated practical improvements to increase the easiness of comprehension, such as simplifying the provided information using pictorial messages and adding local languages to the phone calls and SMS text messages.

### 3.5. Publication Bias

Overall, the funnel plots were asymmetrical, suggesting potential publication bias. However, most studies laid within the non-significant zone of the funnel plot, and no significant outliers were observed; thus, the publication bias did not raise concerns (Appendix A).

## 4. Discussion

### 4.1. Main Findings

Ten RCTs with 3323 stroke patients were identified through systematic literature searches and were included in the quantitative analysis. In summary, the results of the meta-analyses suggested that remote interventions had moderate effects (SMD = 0.49 and 95% CI [0.04, 0.93]) at supporting changes to medication adherence behaviour and clinically significant effects on improving blood pressure (sBP −3.73 mmHg 95% CI [−5.35, −2.10]/dBP MD = −2.16, and 95% CI [−3.09, −1.22]) and cholesterol (MD = −0.36 and 95% CI [−0.52, −0.20]). These results prove that remote interventions have a positive impact on reversing morbidities and mortality and, thus, potentially health care costs associated with stroke. The clinical indicators are the intermediate outcomes for measuring medication adherence changes produced by the targeted interventions; thus, the comparisons of this data are reliable and easily interpretable in usual care practice through standardised units of measurement. For instance, lower blood pressure and cholesterol levels are associated with reduced risks of stroke. Blood pressure, including sBP and dBP, are proven to affect cardiovascular events independently, and thus, changes in both of the indicators were included to improve our understanding of the interventions’ cost-effectiveness [43]. The medication adherence improvements facilitated by the remote interventions found in this review align with the findings of a similar review, which demonstrated that interventions utilising SMS messages and phone calls improved medication adherence in stroke patients [5]. However, before this review there was no evidence on the effective intervention mechanisms of impact.

This study explored the between-study heterogeneity by performing two subgroup analyses to identify the mechanisms that caused the observed variance. The first analysis was based on the COM-B model, suggesting that capability has the biggest effect size, followed by motivation and opportunity. Perhaps, interventions providing higher or tailored dose of support to address Capability, combined with strategies to device Motivation and Opportunity, could be one way to treat medication non-adherence in Stroke populations. Most studies aimed to improve patients’ capability by providing reminders, using a combination of text messages and audiovisual stimuli that were plugged in existing motivations or preferences in order to enhance reflective thinking and proactive planning of the behaviour. The second subgroup analysis compared the self-reported outcomes to the objective measures to determine the potential bias caused by the subjective reporting of the results. No heterogeneity was detected between the groups, although the objective measures had a slightly larger effect size than the self-reported measures. This provides evidence of the robustness of this review, suggesting that there was no subjective bias or over-reporting of the magnitudes and effects of the interventions. 

Publication bias was then assessed using funnel plots, whereby most of the plots appeared somewhat asymmetrical. Publication bias is generally inevitable for systematic reviews, and no significant outliers were found in the plots. Hence, it was concluded that the observed asymmetry does not provide true evidence of the compromised validity of the results.

Following the meta-analysis, the patients’ acceptability of the remote interventions was critically evaluated based on a subsample of the primary studies. The overall feedback on the remote interventions was positive, as all the results suggested high levels of satisfaction and usefulness, as well as the relevance of the information provided. Recommendations suggested that enhanced automated tailoring and personalisation could further improve intervention acceptability.

### 4.2. Clinical Implications

Telemedicine and mHealth interventions have expanded the potential of patient access to medical advice and support from healthcare providers. These interventions aim to facilitate chronic disease management by improving medication adherence and increasing pharmacological treatment success [13]. Previous cost-effective analyses provided evidence regarding the costs associated with non-adherence and suggested that higher drug costs are offset by lower inpatient, outpatient, emergency department, and hospitalisation costs [44,45]. In this review, patients who received remote interventions displayed improved adherence to their prescribed therapy, as well as significant reductions in common stroke risk factors (BP and LDL-C levels), suggesting the potential cost-effectiveness of these interventions. Moreover, these remote interventions received overall positive feedback and high acceptability rates, indicating a higher likelihood for patients to engage with the interventions, leading to increased chances of successful implementation and possibly to improved overall effectiveness of the intervention [46]. 

The results from the subgroup analysis indicated that enhancing patients’ capability was most effective in improving adherence within the COM-B model. This includes improving patients’ comprehension of stroke and its treatments, their cognitive functioning, such as memory and judgement, and their executive cognitive function, such as their capability to plan their treatment [24]. The most effective interventions consisted of medication reminders or information on the risks of stroke and the benefits of its treatments in remote interventions. This suggests that public health efforts should prioritize on improving medication adherence by enhancing the capability of patients with reminder interventions and education to improve knowledge and understanding of the diseases and treatments and bring about clinically meaningful behavioural changes. This finding is consistent with previous reviews that depicted the effectiveness of capability enhancement in increasing adherence, such as providing access to expert advice and social support [47,48]. 

mHealth and telemedicine interventions that target medication adherence are low in cost, convenient, and able to reach a wide and diverse range of target users, enabling equitable access to healthcare on a population scale. This review provided evidence of the effectiveness of the interventions, but further research is required to explore their correlations with tailored adherence rates to predict population health outcomes and inform policymakers to prevent avoidable expenditures on the healthcare systems. 

### 4.3. Strengths and Limitations of This Review

This review has significance in contributing to public health and primary care research as it is the only systematic review that evaluated the effectiveness of integrating telemedicine and mHealth technology into chronic disease management based on the identification of effective intervention mechanisms of action. A thorough search of the literature was conducted in multiple databases, followed by the development of a pre-registered protocol. The references of all the included studies were searched to ensure that the maximum number of eligible studies was included. Furthermore, the data from the eligible studies were evaluated based on the Cochrane guidelines [31], providing an overview assessment of the quality of evidence included in the review. Lastly, the secondary outcome results were strengthened with the dichotomous outcome estimate of the overall effect sizes, as significant improvements in the continuous outcomes of the clinical indicators do not necessarily translate into clinical significance.

One of the major limitations of this review was the limited number of primary studies that met the eligibility criteria. Grey literature was not searched, and most authors that were contacted regarding missing data did not respond. As a result, only ten RCTs were included. This implies that the analyses could potentially be underpowered. The results of the subgroup analyses could be affected by the subgroups that only comprised one or a small number of studies, which was not representative of the magnitude of the intervention effect. In addition, this review was limited by the methods defined in the protocol. Specifically, additional factors, such as the intervention strategies to modify the COM-B components, should be investigated for further exploration of the unexplained within-study heterogeneity to provide insights into the effective mechanisms of applied behavioural interventions. 

### 4.4. Strengths and Limitations of the Included Studies

The included RCTs were based on real-life setting evaluations, which entailed stroke patients using remote interventions in adjunct to usual care and rehabilitation programs. As a result, these trials can be used to synthesise evidence for the development of remote interventions, as well as to inform usual care practice and public health policy to improve the medication adherence of patients with long-term illnesses. The included studies also had varying country settings that spanned across different continents, enabling access to a diverse population that came from different backgrounds. This eliminated concerns regarding inequalities with respect to accessing and using effective remote health interventions.

One limitation of the studies was the small effect size found in the primary dichotomous outcome. This could be caused by the Hawthorne effect, indicating that patients have better adherence than usual due to them being conscious of being monitored [49]. Moreover, as demonstrated in the RoB assessment, the quality of evidence from the primary studies was low. Three out of ten included studies had an overall high risk of bias, which may affect the validity and generalisability of the results in the review. 

## 5. Conclusions

Our meta-analysis demonstrated that remote interventions could effectively improve medication adherence and clinical indicators in stroke patients compared to usual care, potentially providing cost-effective solutions to improve treatment efficacy. Furthermore, high patient satisfaction with the interventions indicated high intervention feasibility and acceptability, suggesting potentially successful implementations of the behavioural strategies. This review also addressed the knowledge gap of the effective mechanisms that impact on improving medication adherence in stroke patients, and thus could inform replicable practices and further intervention developments. Specifically, targeting enhancing capability, combined with supporting motivation and opportunity, is established as the most effective way within the COM-B model to improve adherence behaviours in stroke patients. This review has laid the fundamentals for further research to stratify more specific behaviour change techniques that support treatment adherence to behavioural change. Future studies could utilise more rigorous studies, larger sample sizes, and standardised outcome measurements to improve our understanding of the effective mechanisms of these interventions, as well as bridge the research gaps in the current evidence to inform intervention development and future public health practices.

## Figures and Tables

**Figure 1 behavsci-13-00246-f001:**
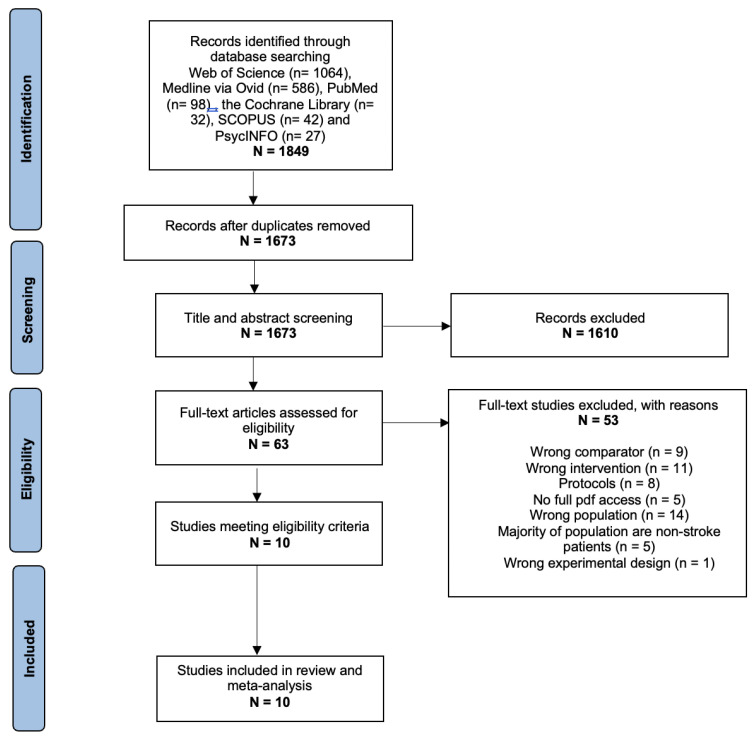
PRISMA flow diagram of systematic search, inclusion, and exclusion of studies adapted from Moher et al. [42].

**Figure 2 behavsci-13-00246-f002:**
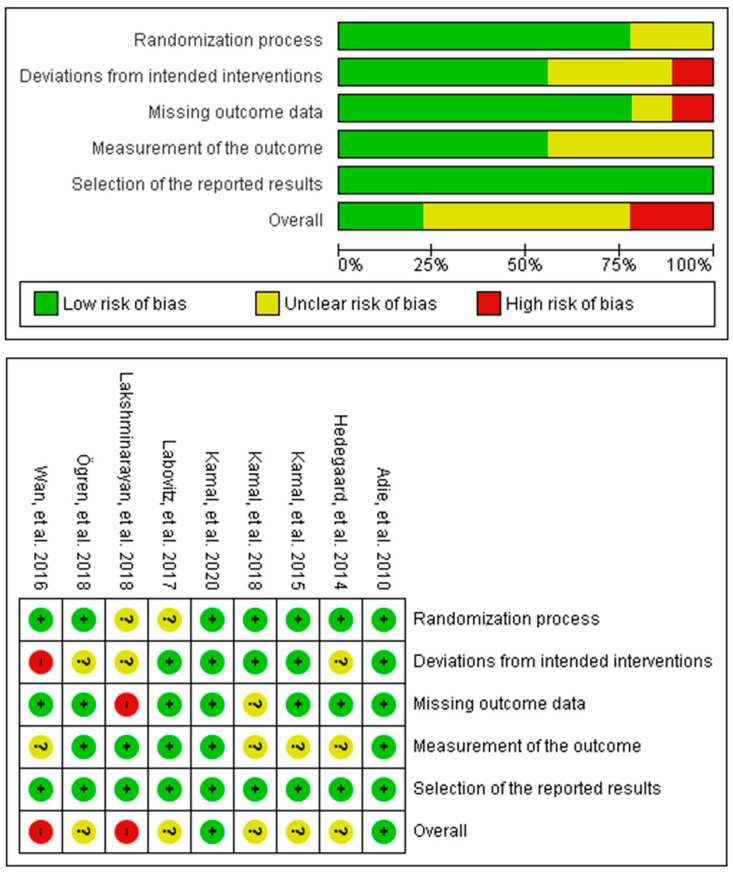
Summary of the overall risk of bias assessment in the individual RCT studies. The cluster RCT was omitted due to the absence of an extra domain (recruitment process).

**Figure 3 behavsci-13-00246-f003:**
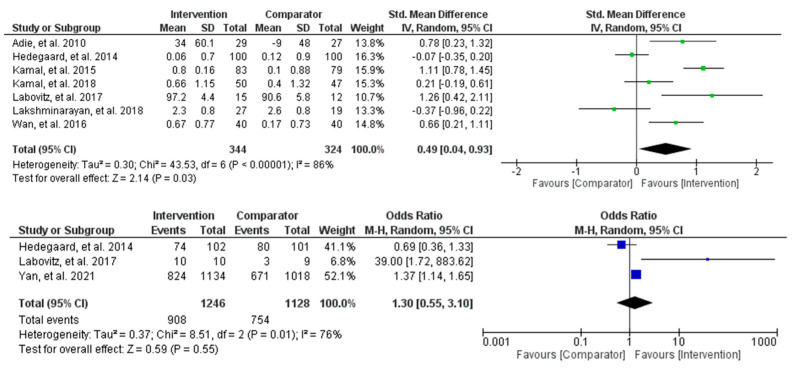
Meta-analysis results comparing changes in medication adherence mean scores and ratios between mHealth, telemedicine, and usual care groups.

**Figure 4 behavsci-13-00246-f004:**
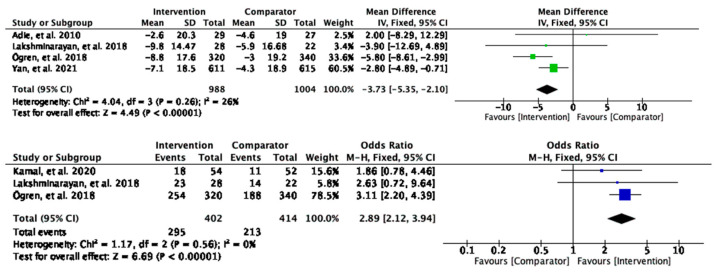
Meta-analysis results of intervention effect on systolic blood pressure.

**Figure 5 behavsci-13-00246-f005:**
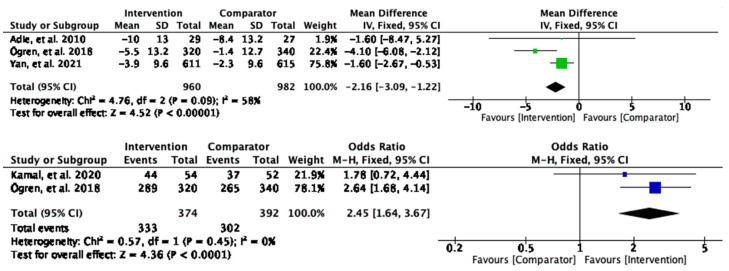
Meta-analysis results of intervention effects on diastolic blood pressure.

**Figure 6 behavsci-13-00246-f006:**
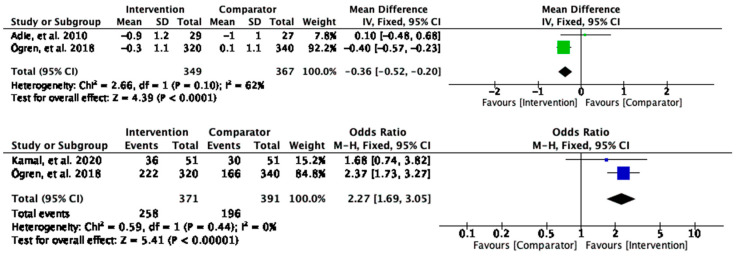
Meta-analysis results of LDL-C outcomes.

## Data Availability

Not applicable.

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
