# Peer review of "Effectiveness of Remote Interventions to Improve Medication Adherence in Patients after Stroke: A Systematic Literature Review and Meta-Analysis"

_behavsci, 2023, doi:10.3390/bs13030246_

Round 1

Reviewer 1 Report

Dear author(s)
It was my pleasure to review your manuscript entitled “Effectiveness of remote interventions to improve medication adherence in patients after stroke: A systematic literature review and meta analysis” and advise you to  prosper your current research project. In my view, your topic has touched on a critical issue in a fascinating context. However, there are many spaces to be improved in terms of argumentation, theoretical background, research method, and findings. I hope my below comments would help you develop your work into groundbreaking research in your domain.
1. The abstract should indicate the innovation of the work.
The introduction should clearly illustrate (1) what we know (the key theoretical perspectives and empirical findings) and what we do not know (major, unaddressed puzzle, controversy, or paradox does the study address, or why it needs to be addressed and why this matters) and (2) what we will learn from the study, and how the study fundamentally changes, challenges, or advances scholars’ understanding. Much sharper problematization is required so that the introduction draws the reader into the paper. At the end of the introduction, we should have a clear idea of what the paper is about (i.e., its motivation, the gap in understanding that the paper is trying to address, and a summary of theoretical contributions).
Paragraph 3 explains what we need to find out.
Paragraph 4 explains briefly what this paper will do to find out, the method, etc.
Paragraph 5, with no references, explains the structure of this paper.
2. In the discussion section, for each case study that you have identified separately, the results should be written, what effects it has on the main result?
3. The conclusion shows the final results of your research (you need a conclusion for your research). Also, the conclusion should be revised to highlight the aims of the study, summarize the finding, and the significance and usefulness of the study.
Please clarify more what are the theoretical and practical contributions of your research.

Best of luck with the further development of the paper.

Reviewer 2 Report

This study reviews the existing literature for telemedicine and mHealth interventions that are aimed at improving medication adherence in stroke patients. The introduction established the need for this systematic review and meta-analysis. The systematic review is conducted using the software Rayyan. The selected studies that meet the study criteria are assessed for risk of bias. The study results are well presented and the discussion is well-written. Minor revisions to improve the manuscript are-

1. In the introduction, please add some statistics on the frequency and number of medications that individuals with Stroke are prescribed on average. Since the study's focus is on medication adherence, this information is important.

2. When 10 studies met the study criteria and were assessed, figure 2 shows the risk of bias for only 9 studies - what about the 10th study that was included? Please provide information regarding the 10th study as well.

3. Provide a reference for the software used for conducting the systematic review.

Round 2

Reviewer 1 Report

Dear authors

Dear author(s)

Hope you are doing well. According to the review of this article, the corrections have been made.

Good luck